# Make My Day: primary prevention of stroke using engaging everyday activities as a mediator of sustainable health – a randomised controlled trial and process evaluation protocol

Ann-Helen Patomella [1,2] Susanne Guidetti [1,3] Maria Hagströmer,[1,2] Christina Birgitta Olsson [1,2] Elin Jakobsson [1] Gunnar H Nilsson [1,2] Elisabet Åkesson [1,4] Eric Asaba [1,4]

¹Department of Neurobiology, Care Sciences and Society, Karolinska Institutet, Stockholm, Sweden
²Academic Primary Health Care Centre, Stockholm, Sweden
³Karolinska University Hospital, Stockholm, Sweden
⁴Stockholms Sjukhem R&D, Stockholm, Sweden

**Correspondence to**
Dr Ann-Helen Patomella;
ann-helen.patomella@ki.se

## ABSTRACT

**Introduction** The individual, societal and economic benefits of stroke prevention are high. Even though most risk factors can be reduced by changes to lifestyle habits, maintaining new and healthy activity patterns has been shown to be challenging.

The aim of the study is to evaluate the impact of an interdisciplinary team-based, mHealth-supported prevention intervention on persons at risk for stroke. The intervention is mediated by engaging everyday activities that promote health. An additional aim is to describe a process evaluation that serves to increase knowledge about how the programme leads to potential change by studying the implementation process and mechanisms of impact.

**Methods and analysis** The study will be a randomised controlled trial including 104 persons at risk for stroke. Persons at risk of stroke (n=52) will be randomised to an mHealth-supported stroke prevention programme. Controls will have ordinary primary healthcare (PHC) services. The 10-week programme will be conducted at PHC clinics, combining group meetings and online resources to support self-management of lifestyle change using engaging everyday activities as a mediator. Primary outcomes are stroke risk, lifestyle habits and participation in health-promoting activities. Assessments will be performed at baseline and at follow-up (11 weeks and 12 months). The effects of the programme will be analysed using inferential statistics. Implementation will be analysed using qualitative and quantitative methods.

**Ethics and dissemination** The study has been approved by the Swedish Ethical Review Authority. Study results will be disseminated in peer-reviewed journals and at regional and international conferences targeting mixed audiences.

**Trial registration number** NCT05279508.

## STRENGTHS AND LIMITATIONS OF THIS STUDY

⇒ A robust randomised controlled trial design will be used to investigate the effectiveness of the intervention programme.
⇒ A process evaluation will provide rich data, useful for analysing the outcomes and the research process.
⇒ A limitation is that primary outcomes are based on self-reported data.
⇒ An inclusion criterion is motivation for change, which can skew the results and lower external validity.

reform targets primary healthcare (PHC) specifically and is designed to proactively address and prevent illness at an early stage to reduce the burden of non-communicable diseases (NCDs), such as stroke. The reform is in line with the United Nations Sustainable Development Goals, urging governments to reduce premature mortality from NCDs by 1/3 by the year 2030 through prevention and treatment.[2] By the age of 55, the risk for stroke increases considerably and doubles each decade afterward.[3 4] Although the incidence of stroke has decreased in the general population, trends show an increase in stroke incidence in young adults.[5] In Sweden, stroke incidence has declined by over 40% in the last 15 years.[6] However, both globally and in Sweden, cerebrovascular diseases such as stroke continue to be the most common cause of death and impairment.[7]

The individual, societal and economic benefits of stroke prevention are high. Many of the stroke risk factors are largely addressable; for example, smoking, obesity, type 2 diabetes, hypertension, physical inactivity and dietary intake. The benefits of a healthy lifestyle are clear[4 8]; however, the long-term effect

## INTRODUCTION

In Sweden, a healthcare reform programme entitled 'Good Quality Local Health Care—A Primary Care Reform'[1] is currently paving the way for a large-scale transformation. The

(follow-up at 12 months or longer) of lifestyle interventions is not.[9] [10] For example, the effectiveness of PHC-based physical activity interventions is inconclusive.[11] There is evidence for short-term improvements, but there is a lack of evidence for long-term effects.[9] A multifactorial approach to stroke prevention is warranted. A systematic review showed that multifactorial lifestyle habit interventions have a greater potential effect on reducing risk factors than single-factor interventions.[12] Preliminary evidence exists from a Swedish trial on a multifactorial lifestyle counselling programme[13] that improved physical activity and dietary habits and reduced smoking and stress; however, the study was not conducted in PHC and did not have a control group. mHealth (a term for the combination of eHealth services and smartphone technology[14]) presents possibilities for accessibility and affordability when developing health services[15] [16] making it an excellent tool for designing stroke prevention models.[17]

### Theoretical concept of the intervention programme

The programme is a complex intervention, and the Medical Research Council (MRC) guidance for developing and evaluating complex interventions[18] has been used in the design of the study. MRC suggests several key elements and stages; development, feasibility/piloting, evaluation and implementation. This randomised controlled trial (RCT) evaluates the effectiveness and implementation of the Make My Day (MMD) programme and is based on already-completed or ongoing studies concerning development and feasibility.

#### Engaging everyday activity

In the MMD programme, engaging everyday activities (EEA) are seen as the means and goal for changing and sustaining a healthy lifestyle in the intervention programme, and while it is not a new concept, it has not been studied in relation to changing lifestyle habits to prevent NCDs. EEA is defined as a special type of activity, that is, based on individual experience, filled with meaning and gives a sense of intense participation and enthusiasm to the individual.[19] [20] Examples from previous studies show that EEA can include a variety of activities such as working, playing computer games and reading books. We recently showed that an intervention programme targeting cardiovascular disease prevention could benefit from incorporating health promoting EEA.[21] The complexity of changing lifestyle habits has been described as a paradox between EEA and health.[22] Even though EEA is subjectively meaningful and engaging to an individual, it might have an arguably negative impact on health; for example, engagement in sedentary activity or in unhealthy behaviours.[22] In the current project, EEAs are seen as having the potential to change everyday activity patterns, and when carefully designed (eg, listening to an audiobook while taking a walk), incorporate a routine that promotes sustainable health among persons at risk for stroke. Studies have shown that promoting EEAs can have positive health impacts

on older adults.[23–25] Studies on populations that live lives that incorporate EEAs combined with moderate-intensity physical activities and a healthy diet indicate a strong relation to well-being, longevity and cultural context.[4] [5]

The current state-of-the-art in stroke prevention suggests the need for a multifactorial PHC intervention that addresses modifiable risk factors for stroke based on individual needs and engagement in everyday activities that promote health with the support of a mHealth service.

### Objectives of the proposed study

The main aim of the study is to evaluate the impact of an interdisciplinary team-based, mHealth-supported prevention intervention in PHC—mediated with EEAs that promote health—to decrease stroke risk (primary outcome), and increase participation in EEAs and health-related quality of life (HRQoL). The aim is also to describe a process evaluation that serves to increase knowledge about how the programme leads to potential change by studying the implementation process and mechanisms of impact.

### Hypothesis

We hypothesise that the MMD intervention programme is more beneficial than regular PHC services (control group) in decreasing stroke risk (primary outcome). We also hypothesised that MMD is more beneficial than regular PHC in increasing (a) participation in health promoting EEAs and (b) HRQoL.

## METHODS AND ANALYSIS
### Design

The study will be a randomised, assessor-blinded, controlled trial of persons at risk for stroke. The process evaluation answers questions as to what interventions were delivered and how by combining qualitative interviews and descriptive quantitative data on the implementation, causal mechanisms and contextual variation.[26]

### Study setting

The study will be conducted in close collaboration with four PHC clinics in the Stockholm area (different parts of Stockholm to represent a diversity in geographical area). PHC clinics in this study are rehabilitation units involving dietitians, physiotherapists and occupational therapists. In Region Stockholm, rehabilitation units are often both organisationally and geographically separate from general practitioner (GP) PHC clinics. These PHC rehabilitation units have an agreement with the County Council in Stockholm and are available for patients to choose from without the need of a referral for treatment by certified physiotherapists, occupational therapists and dieticians. PHC services are publicly funded in Sweden.

### Sample size and power considerations

The primary outcome is stroke risk, with emphasis on modifiable stroke risk factors. Based on data from a case

study,[27] we calculate a decrease of at least one level of stroke risk (eg, moving from high risk to medium risk in the Stroke Risk Score Card[28]) with an SD of 1.5, the statistical power of 80%, with two-tailed $\alpha=0.05$. Under these assumptions, the required sample size was 35 in each group (in total n=70). The Stroke Risk Scorecard was developed as an easy-to-use self-assessment tool by the National Stroke Association in the UK.[28] The tool has been used previously in a few studies to detect risk factors of stroke[27 28] and in a recently finished pilot study conducted in the research group (publication in manuscript). Since the Stroke Risk Scorecard has not been sufficiently tested psychometrically, power calculations were added for performance in EEAs. Assuming a difference on performance in EEAs of two points (as measured with the Canadian Occupational Performance Measure (COPM)[29]) a power of 0.8 and a two-sided p value of 0.05 a sample size of 40 participants in each group would be sufficient.

To safeguard against drop-outs (a maximum 30% drop-out rate is assumed), a total of 104 participants will be enrolled in the study (52 in each group).

## Participants: recruitment and eligibility criteria
Persons at risk of stroke will be included in the project and participants will be recruited via advertisements in social media, a webpage and flyers at PHCs. A stroke risk online screening survey will be used to find eligible participants. Inclusion criteria for the study are (a) three or more risk factors deemed 'high risk' using the Stroke Risk Scorecard, (b) motivation for lifestyle change and to participate in a digital lifestyle intervention (including use of a smartphone) and (c) aged between 55 and 75 years old, and without a diagnosis of dementia or cognitive impairment hindering participation. Exclusion criteria include having previously had a stroke or a transischaemic attack diagnosis, lack of understanding of the Swedish language and not being able to use a mobile phone application. A total sample of n=104 participants (persons at risk of stroke), divided into two arms (52+52) for intervention and controls is estimated. Block randomisation will be used with a block size of 4.[25] Allocation will be done following baseline assessment by a researcher not involved in data collection or intervention. The assessors of outcomes will be blinded to allocation until the end of the study.

In addition, next of kin to persons at risk for stroke in the intervention group are also invited to answer questions (survey and interview) regarding support of their relative. Persons at risk for stroke who do not have a next of kin will not be excluded from the study. PHC professionals who have been trained and delivered the intervention programme will be additional participants in the process evaluation of the study. Stakeholders (such as leaders at the involved PHC clinics) will be invited to individual interviews.

## Informed consent
Written informed consent will be obtained from all participants (persons at risk, their next of kin, and PHC staff and stakeholders) at the start of recruitment. Information about the study will be given in written and verbal forms during meetings with research staff. Persons at risk will be asked to identify a next of kin that will be asked to participate by the researchers.

## MMD: a stroke prevention programme
The MMD intervention programme enables healthy activity patterns and aims to reduce the risk of stroke by means of four strategies: (a) the incorporation of health-promoting EEAs, (b) the use of mobile phone technology (mHealth) to increase health literacy, and awareness of current habits and fostering self-management, (c) setting realistic goals that form new habits that prompt conscious decisions for healthy choices and habits and (d) sharing experience in a learning environment.

## Duration and specific content of the intervention programme
The MMD stroke intervention programme is a 10-week group programme consisting of five sessions over the first 5 weeks, followed by a sixth booster session 5 weeks later. During the intervention, participants will work actively on self-chosen goals, EEAs and habits to change behaviour and lifestyle. A mobile phone app will be used by participants throughout the 10 weeks, supporting their awareness of current lifestyle habits and everyday activities. To form new habits, common situations will be used to cue behaviour change, like seeing an elevator and looking for the staircase, prompting health-promoting behaviour and making a conscious decision to walk the stairs.[30] The continuation of a change process is expected from the participants following the 10-week programme period, and strategies for self-management are anticipated.

Each session (90 min) has a theme and includes some type of activity such as exercise, making a light snack or taking a walk. Group dynamics and personal experience are used to reflect on EEAs in relation to health, doing and future goals. The sessions and content, presented in table 1, are delivered by a trained health professional, for example, an occupational therapist, physiotherapist or dietician. There are sessions weeks 1–5 and week 10. During weeks 6–9, no sessions are held, instead the participants are expected to self-manage these weeks with the support of a mobile phone app.

The healthcare professionals who will provide the MMD programme/intervention will participate in structured education specially designed for this programme. This education will be given in an on-site and digital combination on three occasions and will be held by two research team members with extensive experience in pedagogy and in the research protocol. In addition, the health professionals will have access to a digital educational platform with rich and varied material, and all material to be used during the 10-week programme. To avoid contamination, the health professionals are instructed to not deliver the 10-week programme to other patients during the research period.

**Table 1** Summary of session themes, concepts and activities supporting a change process

| Week | Session theme | Profession | Concepts | Activity |
|------|---------------|------------|----------|----------|
| 1 | 1: Risk factors for stroke and engaging everyday activities | Occupational therapist | Health literacy concerning stroke risk, engaging activities, change process, expectations | Peer interview on engaging activities. Learn how to register in the app |
| 2 | 2: Physical activity | Physiotherapist | Physical activity, physical inactivity | Try a physical group exercise class at a gym |
| 3 | 3: Diet and health | Dietician | Dietary routines and change | Food lab—prepare and test; for example, healthy snacks/sandwiches |
| 4 | 4: Balanced everyday life | Occupational therapist | Activity balance and stressors | Relaxation—such as medical yoga or meditation |
| 5 | 5: The meaning of healthy habits, routines and activity patterns | One of the team members | Current and desired routines/habits, activity patterns and resources | Walk-and-talk—for example, in a forest or a historical walk in the city |
| 10 | 6: Booster session: evaluation and the road ahead | Occupational therapist | Self-management, sustainability, view of the self, social support, revisiting goals/new goals and social aspects of health | Preparing healthy snacks and group reflection on the programme |

## The mobile phone app

The app for the pilot project was produced by collaboration with ScientificMed Tech AB (now part of Cuviva AB) (http://www.scientificmed.com), and for the current project, the app has been modified based on the previous experiences of the users and the researchers. A workshop with the pilot study participants, researchers and the company showed that the participants wanted the app to be more tailored to their needs with a more user-friendly interface. Both the researchers and the users wanted feedback to be relevant and tailored to the users' progress, thus supporting change and awareness. As with the previous version, the new version of the app includes six domains for registering daily activities, experiences and behaviours (see figure 1 for examples from the app): my goals (goal achievements on three preset goals); physical activity and steps (step counts, 24-hour time use in relation to exercise, moderately intense activities, sleep, sedentary activities and other activities); engaging activities (participating in health-promoting EEAs); tobacco and alcohol use (consumption); stress (perceived time-pressure) and dietary habits (consumption of fruits/vegetables, breakfast, fish and snacks—not included in the figure). Domains are based on modifiable risk factors for stroke, as presented by the American Heart Association,[3] with the addition of health-promoting EEAs and stress reduction.

## Data collection
### Persons at risk for stroke

Data collection with a research assistant starts with an individual meeting (baseline) with all eligible participants, just before the meeting (baseline minus 2 days), during which participants are sent a link to an online survey for collecting self-reported measures. During baseline assessment, all participants (including controls) will be informed of their stroke risk factors. Motivational interviewing techniques will be used to identify three problem areas in relation to lifestyle habits and stroke risk factors, and these areas will be used to formulate three lifestyle change goals. Allocation (randomisation) will be done following baseline assessment. Allocation sequences will be done by an independent researcher not involved in data collection or intervention. The researchers who are assessors of outcomes will be blinded to allocation until the end of the study. The assessments measuring primary and secondary outcomes will be collected at baseline, at follow-up (11 weeks) and at 12 months (see table 2). Demographic data will be collected at baseline. Process data will be collected continuously. Controls will be offered standard care by PHCs as needed during the 12-month study period.

## Outcome data

Outcome assessment methods were carefully chosen to assure methods that are valid and reliable and will capture change. The primary outcome measure is risk for stroke, measured by the Swedish version of the Stroke Riskometer[30 31] and the Stroke Risk Scorecard.[27 28] Secondary outcomes include participation in health-promoting everyday activities, measured by COPM,[32] and self-rated health measured using LiSat-11[33] and EQ-5D.[34] Other measures are lifestyle habits (measured using the updated Swedish Lifestyle Survey, Levnadsvaneenkäten),[35] and activity patterns, as measured using the Swedish version of the Daily experiences of Productivity Pleasure and Restoration Profile (PPR).[36 37] Survey data will be gathered for health literacy of stroke risk,[38] experiences of time pressure (stress), cost-effectiveness (eg, self-reported sick leave, healthcare utilisation and use of medication), readiness and motivation for change,[39] current mobile phone use, and mapping out EEAs. Habitual physical activity will be measured using the activPAL

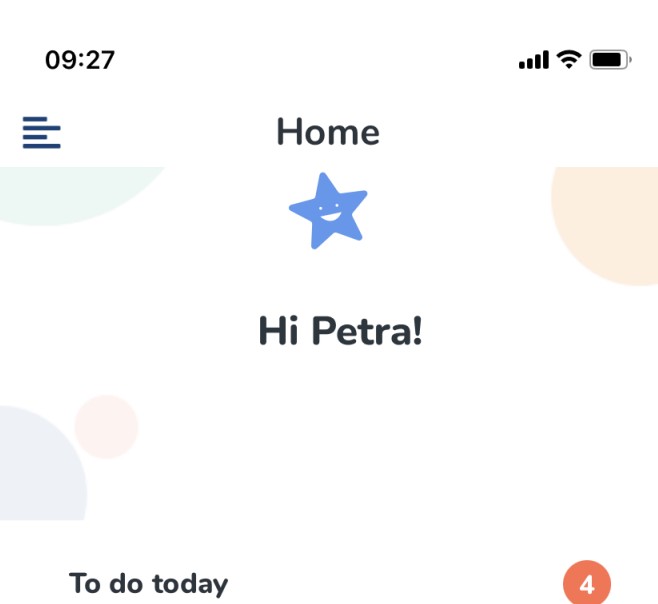

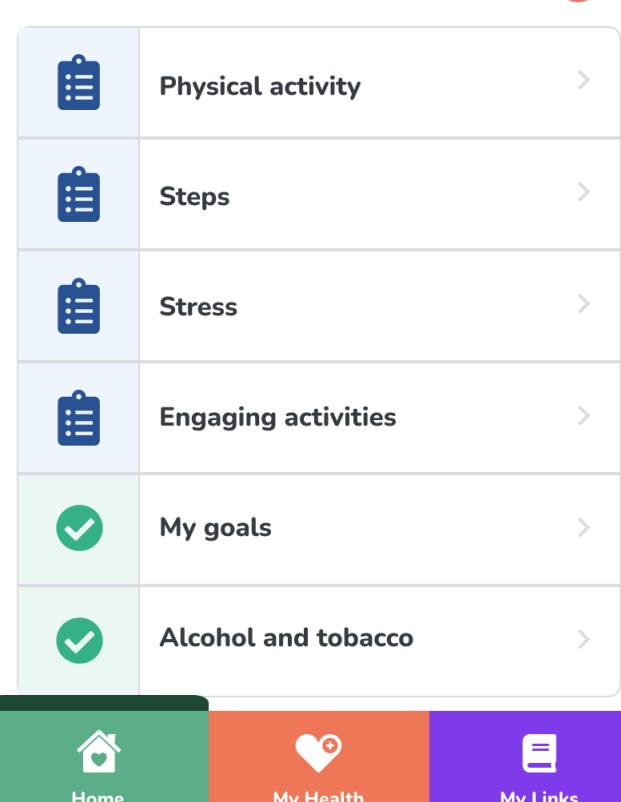

**Figure 1** An example of a checklist that shows the domains that the participants need to register in the app. Published with permission from ScientificMed Tech/Cuviva AB.

micro activity monitor (PAL Technologies, Glasgow, UK).[40] The activPAL is a small device, which provides information on the position and acceleration of the body. The monitor is attached to the thigh and will be worn for five consecutive days after baseline and at follow-up (11 weeks and 12 months). Outcomes from the monitor are (1) time spent sitting/lying, standing, stepping, (2) numbers of step counts and (3) sit-to-stand transitions.

**Table 2** Summary of measures to be collected

| | Instrument and scale | Time points |
|---|---|---|
| **Primary outcome measures from persons at risk for stroke:** | | |
| Stroke risk | The Stroke Riskometer*, the Stroke Risk Scorecard* | t1, t2, t3 |
| **Secondary outcome measures:** | | |
| Participation in everyday activities | COPM, PPR profile* | t1, t2, t3 |
| Physical activity (habitual) | ActivPal | t1, t2, t3 |
| Life satisfaction | EQ-5D*, LiSat-11* | t1, t2, t3 |
| Lifestyle habits | The Swedish Lifestyle Habits Questionnaire* | t1, t2, t3 |
| **Demographics and measures:** | | |
| Age | Year* | t1 |
| Gender | Male/female/other* | t1 |
| Ethnic background | Mother tongue*, place of birth* | t1 |
| Height | Cm | t1 |
| Weight | Kg* | t1, t2, t3 |
| Living situation | Living alone or not* | t1, t3 |
| Yearly income | In Swedish crona* | t1, t3 |
| Employment status | Part time, full time, sick leave, unemployed, student, retired* | t1, t3 |
| Level of education | Years of education* | t1 |
| Blood pressure | mm Hg | t1, t2, t3 |
| Health literacy | Knowledge of stroke | t1, t2, t3 |
| Motivation for change | Self-reported, ordinal scale | t1 |
| Cost-effectiveness | Self-reported sick leave and absence from work past 6 months; healthcare usage past 6 months; Use of medication | t1, t2, t3 |
| **Experiences of next of kin:** | Self-reported health and support* | t2, t3 |
| **Process data:** | | |
| Fidelity and adaptations | Interviews with interventionists on delivery of intervention. Log-books from interventionists. | t2 |
| Dose | Log-books from interventionists. | t2 |

t1=baseline; t2=1 week following intervention ending; t3=12 months follow-up postbaseline.
*Measures collected via an online survey.
COPM, Canadian Occupational Performance Measure; EQ-5D, European Quality of Life 5 Dimensions; PPR, Daily Experiences of Productivity, Pleasure and Resoration Profile.

### Outcome data: next of kin

We will collect data from next of kins to the participants in the intervention group via an online survey. The survey will include demographic measures of health

and questions on their view of the programme and the support they have given their kin during the intervention period.

## Process data
The process evaluation will illuminate causal mechanisms and help identify factors that are associated with variation in outcomes, such as contextual and external factors.[26] Process data include both qualitative and quantitative descriptive data, including logbooks from PHC staff (notes taken during delivery of the programme), course evaluations from the web-based staff training and semistructured exit interviews with participants at risk for stroke and their next of kin (see table 2). Fidelity will be evaluated as the extent to which the programme was delivered as expected. The dose will be assessed as the quantity of the implemented intervention. Adaptation, such as changes made to fit different PHC settings, will be collected during interviews with PHC staff. Reach will be assessed regarding how many eligible patients signed up and how many completed the MMD programme. In addition, adverse events will be registered. Context includes external factors that may act as a barrier or facilitator to the implementation itself and to the interventions' effects. Assessing barriers and facilitators to programme implementation will also involve evaluating programme feasibility; that is, the extent to which stakeholders regard the MMD as satisfactory in terms of content and complexity/difficulty.

Data will be managed using an online software called RedCap (https://www.project-redcap.org/) in combination with a local data management system.

## Participant timeline
Participant enrolment started in April 2022, and in June 2023, all 104 participants had been included. The last groups' 12-month follow-up will occur in March 2024 (marking the end of the study). In total, 5 intervention groups, each consisting of 10–12 participants, will receive the prevention programme during the study.

## Data analysis plan
### Outcomes on effects
The characteristics of all persons at risk for stroke at inclusion, and outcomes at 11 weeks and 12 months after inclusion, will be presented with descriptive statistics. The treatment effects in the RCT study will be analysed on an intention-to-treat basis, with randomised participants retaining their original allocated group, and measured as differences between groups at follow-up and at 12 months considering plausible confounders. Outcome data will be examined for outliers, normality and missing data. Analyses of covariance will be used for continuous outcomes, with baseline values as covariates. Logistic regression analyses will be used for dichotomous outcomes. The level of significance will be set at $p \leq 0.05$ and the confidence level at 95%. We will use the Statistical Package for the Social Sciences (SPSS, Version 29) to analyse the data. Analyses will provide results for the relative effectiveness of the intervention programme. The results will be reported in accordance with the Consolidated Standards of Reporting Trials (CONSORT) recommendations for reporting RCTs on non-pharmacological treatments[41] and the protocol has been reported according to the Standard Protocol Items: Recommendations for Interventional Trials (SPIRIT) reporting guidelines.[42]

## Process evaluation
A mixed-method approach where qualitative and quantitative data are integrated will be used to answer how the implementation process and potential mechanisms of impact can explain the outcomes of the MMD intervention.

Data collected from surveys, logbooks on recruitment and drop-out, and logs from the app registrations will be entered, analysed and summarised. Descriptive statistical analyses will be conducted to report on the study's feasibility: recruitment, drop-outs, retention rate and adherence. Data from app registrations will be used to report on feasibility and usability. Qualitative interviews will be transcribed verbatim and analysed using thematic qualitative analyses.

## Patient and public involvement
The experiences and input from persons at risk for stroke participating in previous studies on the feasibility of the MMD programme have informed the development of research questions, materials and research processes for the current study. The process evaluation will assess the participants' burden of the intervention and time required to participate in the research. We plan to disseminate the study results to all participants in a Swedish report and to ask the participants to comment on the report.

## DISCUSSION
Several NCDs share the same risk factors as stroke, and an intervention programme has the potential to address other NCDs and health in general and should overlap with other health-promoting strategies.[3] In the proposed study, we will evaluate the MMD programme in regard to decreasing stroke risk by using broad strategies and addressing multiple factors of relevance. The theoretical base of the protocol is grounded on EEAs as the mediator and goal for decreasing stroke risk and sustaining personally relevant healthy living habits. It is important to note that the concept of personal relevance can mean that in a total week of different activities, some engaging activities can potentially be considered unhealthy (ie, unhealthy eating activity), but the overall pattern of participation in EEA could be designed to include health-promoting EEA as this study promotes. The paradoxicality of EEAs is that the feeling of being engaged can be just as important for health and well-being as being physically active.[22] Living habits, thus, need to be seen as part of a broader life context, in which health and EEAs are continuously

renegotiated and thus need to be regularly reassessed within the context of each person's life situation.

During the COVID-19 pandemic, there was a strong increase in online PHC consultations in Sweden, especially for younger patients with high economical and educational backgrounds who were born in Sweden; meanwhile, the older population sought less care and preferred face-to-face consultations.[43] Although there is a possibility to deliver the MMD intervention programme completely online (no physical meetings), we have decided to run the programme meetings face to face. During the 12-month follow-up of the pilot study, which occurred at the end of 2020, participants rated physical meet-ups (the possibility of exchanging experiences with other at-risk persons and group leaders) as highly valued, which is in line with previous studies that showed that a blended intervention approach can be efficient compared with only online or on-site intervention.[44] However, whether multiple face-to-face consultations (doses) would be the most efficient is not clear, and is one of the questions for the process evaluation.

The possible limitation of the study will be the reliability of self-reported measures, and there is a risk of bias since reporting might not be accurate, therefore, measures such as activPAL, Body Mass Index (BMI) and blood pressure are complementing the assessments. Although we have planned for 5 days of activPAL wear (the recommendation is 7 days), participants will wear these 24/7, and we will monitor data loss. External validity of the outcomes could be flawed, due to a recruitment process mainly benefiting highly motivated persons at risk of stroke and the risk of drop-outs in less-motivated participants. The power calculations are based on a stroke risk score that has to our knowledge not been used for power calculations previously nor in intervention studies. However, this is the score used in our previous pilot study and most relevant to use, since the aim of the study focuses on modifiable risk factors which are covered in the score. In addition, we have added a power calculation on a secondary outcome.

Ethical dilemmas include that controls are not being supported in the same way as the intervention group and that the recruitment methods could be skewed and fail to reach out to vulnerable groups in society (with lower socioeconomic status [SES]) at risk for stroke. The strength of the study lies in the robustness of the RCT design, the process evaluation and the interprofessional collaboration in a clinical PHC context. The data from the process evaluation will increase and ease the possibility of implementation of a prevention programme for NCDs in PHC. The risk of contamination between control and intervention is deemed minimal, as participants are recruited via social media in a large city.

## Ethics and dissemination

An approval from the Swedish Ethical Review Authority, Sweden has been granted (Ref. numbers. 2015/834-31, 2016/2203–32, 2019/01444 and 2021-05902-02). Data management will be complying with the general data

protection regulation (GDPR), and all data will be stored securely to protect the confidentiality. Participation in the study is not expected to lead to health risks or complications, and potential health consequences will be monitored. Participants who experience any health-related problems during the study will be guided to contact their GP. Participants may choose to interrupt their participation in the study at any time. Researchers can also discontinue a participant's participation based on health issues, or reasons that might jeopardise that person's safety. Reasons for interruption will be recorded. For a summary of the consent form, see online supplemental file 1.

The findings of the study will be published in peer-reviewed journals, and the results will be disseminated to participants, the public, PHC staff and decision-makers through national and international conferences, as well as study-specific web pages.

**Contributors** A-HP, EAsaba and SG conceived the original idea and outline of the study. A-HP is implementing the protocol in primary healthcare settings, with regular dialogue and review by CBO, EÅkesson, GHN, EAsaba and SG. CBO, MH, GHN and EÅkesson contributed to the design of the study. A-HP wrote the study protocol, together with EAsaba and EJ. All authors discussed and commented on draft versions and approved the final version.

**Funding** This work was funded by grants from FORTE, grant number 2020-00175.

**Disclaimer** This funding source had no role in the design of this study and will not have any role during its execution, analyses, interpretation of the data, or decision to submit results.

**Competing interests** None declared.

**Patient and public involvement** Patients and/or the public were not involved in the design, or conduct, or reporting, or dissemination plans of this research.

**Patient consent for publication** Consent obtained directly from patient(s).

**Provenance and peer review** Not commissioned; externally peer reviewed.

**ORCID iDs**
Ann-Helen Patomella http://orcid.org/0000-0003-2667-4073
Susanne Guidetti http://orcid.org/0000-0001-6878-6394
Christina Birgitta Olsson http://orcid.org/0000-0003-1300-7765
Elin Jakobsson http://orcid.org/0000-0001-9756-1354
Gunnar H Nilsson http://orcid.org/0000-0001-7811-7602
Elisabet Åkesson http://orcid.org/0000-0002-8227-9118
Eric Asaba http://orcid.org/0000-0002-6910-3468

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
