## [Reviewer comments · BMJ Open]

ARTICLE DETAILS

TITLE (PROVISIONAL)	Make My Day: Primary prevention of stroke using engaging everyday activities as a mediator of sustainable health—a randomised controlled trial and process evaluation protocol
AUTHORS	Patomella, Ann-Helen; Guidetti, Susanne; Hagströmer, Maria; Olsson, Christina; Jakobsson, E; Nilsson, Gunnar; Åkesson, Elisabet; Asaba, Eric

VERSION 1 – REVIEW

REVIEWER	Guan, Qiongfeng University of Chinese Academy of Sciences Huabei Hospital
REVIEW RETURNED	04-Mar-2023

GENERAL COMMENTS	This is really an interesting research. With the help of modern technology, it is wonderful to help the whole communities to carry out primary stroke prevention intervention. I have a question that there are many causes of stroke. Common causes such as smoking, hypertension and obesity, some can indeed be reduced the risk through education and intervention. However, there are some causes, such as cerebral aneurysm, arteriovenous malformation, atrial fibrillation, cardiac valve disease, amyloidosis, syphilitic vasculitis and other special causes. By random method, a total of 52 cases in each group, several special causes will have a great impact on the final results, etiological stratification , bias correction and statistical analysis is a challenge.
--

REVIEWER	Heron, Neil Queen's University Belfast, General Practice
REVIEW RETURNED	31-Mar-2023

GENERAL COMMENTS	I like the paper and concept of the study overall - well done. My main concern is around the primary outcome - the stroke risk score. This isn't something that is used outside Sweden, to the best of my knowledge. Why don't you use another primary outcome measure, such as blood pressure, that is clearly related to stroke risk. For example, what about a 5mmHg reduction in systolic blood pressure and proving this at 3 months of follow up? Also, I didn't see a power calculation (apologises if I missed it). This needs to be done and clearly laid out, to ensure the sample size is appropriate and well justified.
--

REVIEWER	Zhu, Luwen Heilongjiang University of Chinese Medicine, Fourth Affiliated Hospital
-----------------	---

REVIEW RETURNED	13-May-2023
-------------

GENERAL COMMENTS	First of all, I must say that the logic and expression of this article are not very clear, and even people have no desire to read it. This is the biggest problem, as it leads to details that we cannot understand the authors, such as the number of people included in the studies and the grouping section, and hopefully the authors will express themselves more clearly rather than simply make up the numbers.
--

REVIEWER	Shen, Ying Jiangsu Province Hospital and Nanjing Medical University First Affiliated Hospital, Rehabilitation Medicine Cente
-----------------	---

REVIEW RETURNED	24-May-2023
-------------

GENERAL COMMENTS	In this protocol, the authors seek to evaluate the impact of an interdisciplinary team-based, mHealth-supported prevention intervention in primary health care (PHC). The implementation of this study may provide a new approach to primary prevention of stroke risk and even other NCDs share the same risk factors as stroke. The experimental design is somewhat innovative and rigorous and the contents are described in details, However, there are some contents that needs to be improved. Major comments:  1.The description of the purpose in the abstract does not mention stroke patients at all; can stroke patients be representative of all recipients of PHC? 2.Are there definitions of long- and short-term effects in the impact of lifestyle interventions? What is the specific duration of previous studies on short-term effects? 3.The “Participant: Recruitment and eligibility criteria” section mentions the need for the next of kin to persons at risk for stroke to answer questions about support of their own health, does this affect the subject themselves? 4.The “Outcome data- next of kin” section states how the "next of kin to participants in the intervention group" data will be collected. What about the data from the next of kin in the control group? From the previous description, I understood that the study required the participation of the next of kin to all participants. 5.The measures of “living situation, yearly income, employment status” will be collected at t1 and t3 in the Table2. Why did the authors choose these two points in time? Minor comments:  1.Should the "week" column in the last row of Table 1 be 6-10? 2.The Line 1 of the part “The mobile phone app”: is "by" missing between "produced" and "collaboration"? And underneath this paragraph it says "Insert Figure 1 here", but the Figure 1 is placed at the end of the protocol. 3.The first subheading in the “Data collection” section: the spelling of “stoke” is wrong. And under this subheading it says “just before the meeting (T minus 2 days)”, does the "T" here refer to "t1" in the table 2? 4.The line1 in the “Participant timeline” section: the spelling of “enrolment” is wrong.
--

VERSION 1 – AUTHOR RESPONSE

Reviewer: 1 Dr. Qiongfeng Guan, University of Chinese Academy of Sciences Huabei Hospital	
Comments and questions	Answers and justifications
1. This is really an interesting research. With the help of modern technology, it is wonderful to help the whole communities to carry out primary stroke prevention intervention. I have a question that there are many causes of stroke. Common causes such as smoking, hypertension and obesity, some can indeed be reduced the risk through education and intervention. However, there are some causes, such as cerebral aneurysm, arteriovenous malformation, atrial fibrillation, cardiac valve disease, amyloidosis, syphilitic vasculitis and other special causes. By random method, a total of 52 cases in each group, several special causes will have a great impact on the final results, etiological stratification, bias correction and statistical analysis is a challenge.	Thank you for the comments. We are aware the causes of stroke are many, in the study protocol we describe a study that addresses modifiable stroke risk factors, factors that can reduce with lifestyle intervention. You are correct that there can be several other causes of stroke. In the current study, our outcome is stroke risk (not stroke incidence) and we will follow (evaluate) the modifiable risk factors carefully. We have added text in the Discussion section where we highlight the lack of a validated assessment on stroke risk factors that include modifiable factors, factors that are central in this study.
Reviewer: 2 Dr. Neil Heron, Queen's University Belfast, Keele University	
Comments and questions	Answers and justifications
1. My main concern is around the primary outcome - the stroke risk score. This isn't something that is used outside Sweden, to the best of my knowledge. Why don't you use another primary outcome measure, such as blood pressure, that is clearly related to stroke risk. For example, what about a 5mmHg reduction in systolic blood pressure and proving this at 3 months of follow up?	We are using the stroke riskometer that have been developed in an international collaboration, see references. We are specifically using the Swedish version as there is a translated version available. The stroke riskometer include several non-modifiable and modifiable stroke risk factors. We will also collect data on blood pressure at all time point see Table 2. Thank you for the example on blood pressure in mmHg.

2. Also, I didn't see a power calculation (apologises if I missed it). This needs to be done and clearly laid out, to ensure the sample size is appropriate and well justified.	A power calculation can be found in the method section under the heading Sample size and power considerations. We used two different assessments to calculate power and sample size, however, this can be a limitation and we have discussed this further in Discussion (new text).
Reviewer: 3 Dr. Luwen Zhu, Heilongjiang University of Chinese Medicine	
Comments and questions	Answers and justifications
1. First of all, I must say that the logic and expression of this article are not very clear, and even people have no desire to read it. This is the biggest problem, as it leads to details that we cannot understand the authors, such as the number of people included in the studies and the grouping section, and hopefully the authors will express themselves more clearly rather than simply make up the numbers.	Thank you for pointing that out. We have now worked on the text and clarified the part about the number of people included and the grouping section. We also have discussed the limitations that we can see with using a stroke risk score that has not been very widely spread yet.
Reviewer: 4 Prof. Ying Shen, Jiangsu Province Hospital and Nanjing Medical University First Affiliated Hospital	
Comments and questions	Answers and justifications
Comment: In this protocol, the authors seek to evaluate the impact of an interdisciplinary team-based, mHealth-supported prevention intervention in primary health care (PHC). The implementation of this study may provide a new approach to primary prevention of stroke risk and even other NCDs share the same risk factors as stroke. The experimental design is somewhat innovative and rigorous and the contents are described in details, However, there are some contents that needs to be improved.	Many thanks for the thorough reading and relevant questions that we believe improved the manuscript.

1. The description of the purpose in the abstract does not mention stroke patients at all; can stroke patients be representative of all recipients of PHC?	Thank you for observing this, we have now added the population in focus to the aim.
2. Are there definitions of long- and short-term effects in the impact of lifestyle interventions? What is the specific duration of previous studies on short-term effects?	The studies used in the background define long-term effects as 12 months or longer, but no real definition of short term. We have revised the manuscript to make the definition explicit.
3. The "Participant: Recruitment and eligibility criteria" section mentions the need for the next of kin to persons at risk for stroke to answer questions about support of their own health, does this affect the subject themselves?	We realize the sentence was not well written and have rewritten accordingly.
4. The "Outcome data- next of kin" section states how the "next of kin to participants in the intervention group" data will be collected. What about the data from the next of kin in the control group? From the previous description, I understood that the study required the participation of the next of kin to all participants.	This data is part of the process evaluation to help us better to understand the context that the intervention group is doing their change within. We see that support from next of kin could potentially be an important factor for change. No data from next of kin in the control group will be collected. We have clarified this in the text.
5. The measures of "living situation, yearly income, employment status" will be collected at t1 and t3 in the Table2. Why did the authors choose these two points in time?	This is correct and was decided as we did not expect these measures to change in 12 weeks between baseline and follow-up, but could potentially change at 12 months.
6. Should the "week" column in the last row of Table 1 be 6-10?	No, there is no sessions during week 6-9, we have clarified this in the text above the table.
7. The Line 1 of the part "The mobile phone app": is "by" missing between "produced" and "collaboration"? And underneath this paragraph it says "Insert Figure 1 here", but the Figure 1 is placed at the end of the protocol.	Thank you for noticing this, it has been added. The Figure is in a different format, that is why it is placed at the end. Sorry for the inconvenience.
8. The first subheading in the "Data collection" section: the spelling of "stroke" is wrong. And under this subheading it says "just before the meeting (T minus 2 days)", does the "T" here refer to "t1" in the table 2?	We have taken care of the spelling error. We changed the T to baseline to clarify.
9. The line1 in the "Participant timeline" section: the spelling of "enrolment" is wrong.	Many thanks, we and our language reviewer did not see these errors.

VERSION 2 – REVIEW

REVIEWER	Heron, Neil Queen's University Belfast, General Practice
REVIEW RETURNED	19-Jul-2023

GENERAL COMMENTS	Well done on putting this research together - I look forward to seeing the results and future publications.
---

REVIEWER	Shen, Ying Jiangsu Province Hospital and Nanjing Medical University First Affiliated Hospital, Rehabilitation Medicine Centre
REVIEW RETURNED	27-Sep-2023

GENERAL COMMENTS	The author has revised the manuscript carefully.
--